# Understanding DOHaD Concepts Among New Zealand Adolescents: A Qualitative Exploration of Knowledge, Intervention Windows, and Information Accessibility

**DOI:** 10.3390/ijerph21121556

**Published:** 2024-11-25

**Authors:** Melenaite Tohi, Siobhan Tu’akoi, Mark H. Vickers

**Affiliations:** 1Liggins Institute, University of Auckland, Auckland 1023, New Zealand; melenaite.tohi@auckland.ac.nz; 2Maurice Wilkins Centre, University of Auckland, Auckland 1023, New Zealand; 3School of Population Health, University of Auckland, Auckland 1023, New Zealand; s.tuakoi@auckland.ac.nz

**Keywords:** DOHaD, first 1000 days, non-communicable disease, adolescence, adolescents, health promotion, health literacy

## Abstract

The Developmental Origins of Health and Disease (DOHaD) framework has highlighted the role of maternal and paternal health on disease risk in offspring and across generations. Although adolescence is increasingly recognised as a key DOHaD window where interventions may have the greatest impact in breaking the cycle of non-communicable diseases, data around the recognition of this concept in adolescents remain limited. Previous work by our group found that the understanding of DOHaD-related concepts among adolescents in New Zealand was low, including some adolescents showing disagreement with key DOHaD concepts. This qualitative study aimed to explore DOHaD perspectives and understandings among a group of adolescents who responded to the survey using semi-structured focus groups and interviews (*n* = 12). Four core themes were identified: 1. knowledge of DOHaD and DOHaD-related terminology; 2. understanding different life course windows for DOHaD interventions; 3. recognising that DOHaD-related information needs to be accessible for adolescents; and 4. the importance of developing context-specific resources for adolescents. Adolescents in this study indicated that they had not heard of DOHaD or related terminology. Although the majority recognised that there were many important life stages for potential interventions, there was a strong emphasis on adolescence as a key window of opportunity. Adolescents suggested that more could be done in schools to help promote awareness and understanding of DOHaD-related concepts during the later years of schooling. The development of future resources needs to be contextually specific for adolescents to ensure increased uptake of information during this important developmental window.

## 1. Introduction

Research in the field of the Developmental Origins of Health and Disease (DOHaD) continues to emphasise the importance of relationships between environmental influences in early life and health outcomes later in life. Evidence of the association between poor maternal nutrition during pre-conception and pregnancy and increased risk of non-communicable diseases (NCDs) for the offspring and across generations has been well established [1,2]. More recently, the importance of paternal nutrition on the programming of later disease risk in offspring has also been recognised [3]. Although both parents’ contribution to the overall health of the offspring has been focused on in previous research, the period of adolescence is also increasingly recognised as a key developmental window [4,5,6,7] whereby increased DOHaD awareness at this age can contribute to the development of healthy behaviours that reduce risk factors for later life disease. Literature shows that health promotion can be most effective when the focus is centred on engaging the affected communities in research and intervention development, such as with populations that experience a high prevalence of NCDs [8]. The increasing evidence suggesting low levels of DOHaD awareness in adolescents [9,10,11,12,13,14] necessitates a call for research in this field to include the translation of evidence into practice through health promotion during this key life course period [15].

New Zealand (NZ) has the third highest adult obesity rate in the OECD with over a third of individuals over the age of 15 years classified as obese [16]. There is also disparity in overweight/obesity prevalence rates based on ethnicity with more than 70% of Pacific people affected by obesity compared to 51% in Māori and just over 31% in European or other ethnic groups [17]. This trend is also present in younger age groups with over 10% of children classified as obese with more than 35% of Pacific children living with obesity [17]. Health promotion and knowledge translation related to DOHaD and early-life nutrition has been undertaken with adolescents across many different countries [10,13,18,19,20] including NZ [21]. A 2017 study in NZ schools highlighted that engagement with 11-14-year-olds to improve health literacy and understanding of basic DOHaD concepts resulted in positive behavioural changes; however, these adolescents were only followed through to 6 months after intervention [9]. Likewise, a study in the United Kingdom showed positive behavioural changes in adolescents following an intervention to improve knowledge around DOHaD concepts, but that these trends were not sustained beyond 12 months [11].

Increasing awareness of DOHaD concepts during adolescence, a period where individuals gain increasing freedom and agency to make lifestyle choices, can contribute to reducing the burden of disease [22]. However, despite the high prevalence rates of obesity and related NCD risk factors, and the potential for DOHaD awareness to contribute to prevention strategies and breaking the cycle of disease, there remains a paucity of data that investigates NZ adolescents’ understanding of DOHaD concepts.

A standardised questionnaire, which examines public understanding of DOHaD concepts [21,23,24], was recently adapted by our group to fit the NZ context, with a focus on Pacific people. This survey was used to gather information on baseline knowledge of DOHaD in adolescents living in NZ. Overall, the understanding of DOHaD concepts was low, with some adolescents showing disagreement with some DOHaD-related statements [25]. In order to discuss these trends and explore adolescent views more thoroughly, focus groups and interviews were conducted with a group of survey respondents. The following objectives guided focus groups and interview discussions:To explore current knowledge of DOHaD and DOHaD-related terminology.To understand reasons why some adolescents disagreed with certain DOHaD-related concepts introduced in the recent NZ quantitative survey.To gather adolescent perspectives of current health promotion strategies and identify ways DOHaD researchers and health promoters in this field can support knowledge and understanding among adolescents living in NZ.

## 2. Materials and Methods

### 2.1. Recruitment and Study Design

This qualitative study utilised a semi-structured approach through a combination of focus groups and interviews. All adolescents aged 16 to 19 years old who took part in the previous quantitative DOHaD survey [25] were provided an opportunity to indicate their interest in participating in future research at the end of the survey. Invitations for this qualitative study were therefore sent out to all those who indicated positively. Ethics approval was obtained from the University of Auckland Human Participants Ethics Committee (reference number: 25604). The process of obtaining informed consent for this study was designed to ensure ethical compliance and protect the rights of all participants. Prior to participation, detailed information sheets outlining the study’s objectives, procedures, potential risks, and benefits were provided to the adolescent participants. These documents were crafted in clear, accessible language to ensure comprehensive understanding. Subsequently, researchers were available to provide more information or answer any questions or concerns that the adolescent participants had. Both written and verbal consent was obtained from the adolescents, underscoring their voluntary participation. Consent from the guardians was not required as all the adolescent participants were 16 years and older. Focus groups and interview sessions were conducted during December 2023 until saturation of themes was reached. Each session took place via Zoom (with audio recording) at a time that was suitable for participants. Additionally, measures were implemented to maintain confidentiality and anonymity, ensuring the protection of participant identities throughout the research. Discussion questions focused on health issues that were important to adolescents, their understanding of key DOHaD concepts, and explored perceptions of key results from the quantitative DOHaD survey.

### 2.2. Data Collection

Convenience sampling was used to engage a total of 12 participants across two focus groups and three one-on-one interview sessions. This non-probabilistic sampling method [26] was used to recruit participants based on their willingness and availability to participate in the research. Table 1 outlines the characteristics of the participants who took part. All were aged between 16 and 18 years and included eight females and four males. Seven of the 12 identified with a Pacific ethnicity and 5 identified as European. A majority of the participants were high school students (10), alongside 1 currently employed and another enrolled as a university undergraduate student. Participants primarily resided in Auckland (*n* = 9) alongside one each from Wellington, Christchurch, and Otago.

To ensure consistency in the process, an initial set of open-ended questions was formulated to provide the foundation of the discussions. Discussions focused on the different questions in the survey, investigated the reasons why adolescents might disagree with key DOHaD-related concepts, and explored ways in which DOHaD information could be presented to be both culturally and contextually relevant to adolescents. Participants also had the opportunity to introduce other related topics they felt were relevant to their age group, which were then incorporated in the questions for the following sessions. All focus group and interview sessions lasted approximately 40 min.

### 2.3. Data Analysis

Discussions were audio-recorded, transcribed, and thematically analysed to identify patterns of meaning within the data. The six-phase process identified by Braun and Clarke was used and involved data familiarisation, coding, searching for themes, review and refining themes, defining the themes, and reporting in NVivo [27]. An inductive approach centred on the philosophical framework of critical realism was used to ensure that the development of themes was directed by the data [28]. This approach was chosen to reduce the possibility of researcher bias influencing the interpretation of data, while acknowledging that the research team carries this task with the knowledge of the existing literature [29]. Interpretivism was the chosen epistemological approach to interpreting meanings from the in-depth discussions with the participants of this study [30]. This process allowed for themes that were outside the original focus group questions prepared for the interviews. To ensure reliability of the data and reduction in potential bias, transcribed data were independently coded and then checked for consistency by two authors (S.T. and M.T). Validation of the themes were obtained through consultations with the research team and checked by M.H.V. Each participant agreed to the use of codes that protected their personal identity for reporting purposes.

## 3. Results

Discussions with adolescents revealed four overarching themes: knowledge of DOHaD-related terminology, understanding of the different life course windows for DOHaD interventions, recognition of the need for DOHaD information to be made accessible, and the importance of developing context-specific resources for adolescents.

### 3.1. Knowledge of DOHaD-Related Terminology

All participants in this study identified that they had not heard of the term “Developmental Origins of Health and Disease”, while only two were able to recognise the term “First 1000 Days”. One participant discussed, “I think I have heard of it… isn’t it just the first 1000 Days of life is the most important or something like that?” (Female, 18 years). When questioned about the term “non-communicable diseases”, all participants indicated that they did not know what it meant. Participants pointed out that it may be unrealistic to expect knowledge of these terms as many adolescents do not use such formal ways of communicating. They suggested that complex terms need to be rephrased in order to aid understanding among adolescents.

Non-communicable diseases is not a word I or my family use on a daily (Male, Pacific, 18 years).We were born into homes where we don’t even know what adolescence means or non-communicable diseases. Until someone explains what it means then I would catch on and be able to give examples (Male, Pacific, 18 years).

Despite not recognising the term NCDs, adolescents were able to discuss factors they thought may contribute to someone being at risk of developing such diseases, for example, genetics, lifestyle, and not having access to healthy food. A participant also recognised the influence of broader socio-economic factors and how these can influence what people have access to.

Genes, obesity (and) exposure to other people or like who you are surrounded by (Male, Pacific, 18 years).Certain communities may not have as good kind of health outcomes in certain instances due to things I guess like institutionalised racism… In terms of healthy eating and stuff especially with the cost of living, fresh fruit and vegetables are like super expensive… sometimes people want to make like healthier choices, but they just can’t because society isn’t helping them in that way (Female, European, 17 years).

The disconnect between adolescents and DOHaD-related terminology was also evident in the participants’ prioritisation of health issues for their age group. Figure 1 shows that two of the most important health issues highlighted by adolescents as being relevant to them were alcohol abuse and drug addiction. The participants chose these two health issues because they explained that these are the most well-known health issues and that they often hear about their impacts on health whether at school, social media, or in the wider community.

### 3.2. Understanding Different Life Course Windows for DOHaD Interventions

In the discussions with participants, they identified different age groups they perceived as being crucial for health. While the majority of participants identified that they believed all age groups were important, a strong emphasis was placed on younger generations, such as adolescents. Participants discussed that during this period, one’s health behaviour affects the rest of their life and noted how it can be challenging to make lifestyle changes later in adulthood. Only one participant shared different views, specifying how they believed both infants and elderly people were two critical age groups for health interventions.

I guess everyone like at every kind of stage at life (is important) … definitely with younger age groups you can set good foundations for healthier choices (Female, European, 17 years).All (are important). But I think most important is 0–18 years while you are developing. Because that is when majority of your development happens… when you’re a bit older you become more stubborn, and you’re stuck in your ways (Female, European, 18 years).Infants are small, and they just came into the world, so their immune systems are not as strong as teenagers and adults… elderly are kind of like infants as well. As they grow, their body gets weaker (Female, Pacific, 18 years).

Although agreeing that the period of adolescence was important for later health, participants in this study signalled that this knowledge may be lacking, particularly in relation to how it impacts the next generation. They discussed that medical practitioners, schools, parents, caregivers, and other adolescents were all key groups who should understand the importance of the adolescent period for lifelong health. These suggestions were based on their perception of who could be trusted to communicate the facts with them regarding the importance of the adolescent window.

Medical practitioners (are important) because sometimes you might have an issue and they do not know much about how that issue could differ between ages. (Being educated in DOHaD) might make them kind of understand us better and be able to respond to issues better (Female, European, 18 years).Parents should know what is going on with our health and wellbeing because they birthed us and look after us… our families are a big part of our decision making when it comes to health (Female, Pacific, 16 years).I would say young people themselves (are important), that way you kind of gained a little bit of independence in your thinking and in your actions so it can be liberating in a way. But I think also a close second would have to be the family and the parents because it is a lot easier to train those good habits when the child is very young. It gets more difficult as they age (Female, European, 18 years).

Focus group and interview discussions also explored adolescent views on trends that resulted from the quantitative survey that was undertaken. Table 2 outlines three key DOHaD statements from the survey (the possible responses ranged from “strongly agree” to “strongly disagree”) that adolescent respondents predominantly disagreed with. Discussions explored potential reasons why the participants in this study thought their peers might have disagreed with such statements.

Seventy-five percent of participants highlighted that disagreements with statement (i) (the mother’s health before she becomes pregnant affects the health of the foetus) could be due to misconceptions and a lack of knowledge among adolescents. Table 2 shows example quotes from adolescents who reason that the health and lifestyle of a mother before she became pregnant could not have any impact on the future foetus, as in their view, “you do not exist” yet. Other participants talked about how public dialogue typically focuses on what women should and should not do during pregnancy, rather than emphasising the role of both parents during the period before pregnancy, and this might influence what adolescents perceive as being important.

Discussing why many survey respondents disagreed with statement (ii) in Table 2, regarding whether the father’s health before their partner becomes pregnant affects the health of a foetus, adolescents highlighted that this could be because the father is not physically involved in the carrying of the baby during pregnancy. One participant discussed that since the mother has “closer contact to the baby” they expected that her health would therefore be more important than the health of a father. In terms of how a foetus may be impacted this way, five participants discussed factors such as a lack of support or illness of a partner that could in turn cause stress for the mother and foetus. Only one participant recognised that a child carries DNA from both parents and thus the health of the father can affect a future child.

Disagreements with statement (iii) regarding if a child’s health during childhood and adulthood is influenced by what they are fed during the first two years of life, was explained by participants in relation to the agency people have to make healthy choices later in life (see Table 2). Participants discussed that life is long and that there is time to make lifestyle changes that can negate any adverse experiences in the first two years of life. Adolescents also emphasised the importance of cultural and environmental factors in playing a leading role for shaping how healthy adolescents are now and in generations to come.

If you (adolescents) are trying to eat healthy food and stuff, that is now your way of life and when you conceive or bring children to this world, you will pass on that culture of your life. That kind of affects how they (future children) eat too (Male, Pacific, 18 years).Being a parent, you do influence what your children eat… the baby’s health is shaped both when the baby is inside of the mum and when it is out in the world. So, I would say it is both (Female, Pacific, 18 years).

### 3.3. Recognising That DOHaD-Related Information Needs to Be Accessible for Adolescents

Adolescents discussed that they typically receive most of their health information from social media, Google searches, television news, documentaries, and other online resources. Some noted that their schools did have some resources on health topics but were only made available if asked for by the students. Regarding what is taught within school classes, participants discussed that they received some information on healthy choices in their early high school years, but that this was not continued in the latter years.

I guess TV news and documentary is my primary source. I end up getting recommended content from a lot of doctors and medical professionals on Instagram, YouTube and other platforms where they talk about things which help debunk health myths and misconceptions (Female, European, 17 years).I have kind of experience (with) a lot more messaging towards healthier choices and lifestyle things when I was kind of a bit younger. Because like in my school health class, it was only compulsory for like years 9 and 10 [i.e., the first two years of high school] where we were able to kind of get messaging and learning about healthy choices. But then in the later three years where people are more likely to do more risky things or things that impact their health, that messaging kinds of just goes away (Female, European, 17 years).

Adolescents agreed that DOHaD-related information, which signifies the importance of adolescent health for future generations, should be promoted consistently throughout high school. They discussed that regular exposure to this information would aid retention and help adolescents make more informed decisions about their health as they grow older. Additionally, adolescents also acknowledged that the home setting could be considered one’s first “school” and thus could be a significant place to discuss such issues.

You don’t retain much information if you are not reminded constantly. You only retain the highlights. We did a whole internal about drugs. What I remember is that drugs are bad but nothing in details (Female, European, 18 years).I think it should be talked more about in school. We have health and PE (physical education) classes but we all know oh exercise is good, eating healthy is good but we do not know why it is so important to do all these good habits when we are younger. Why would you do something if you do not know what you are doing it for? We (adolescents) know it is to be healthy but what is the real reason? Why is it important? (Female, European, 18 years).I think what is most important is the parents. They can talk the child through in a way that the child can understand. Even in school, a lot of students do not care about health, they are just wanting to go home.I think teaching should come from the family (Female, European, 18 years).

### 3.4. The Importance of Developing Context-Specific Resources for Adolescents

Participants in this study reiterated their perception that adolescents are eager to learn and have the potentials to positively impact their health outcomes in the future. They discussed that they yearn to be recognised, heard, and understood by adults when it comes to taking control of their own health.

I feel like we are adolescents, we are not taken seriously. Adults don’t necessarily think of us as equal. Because we are younger, adults think we do not know a lot. They usually think that we won’t understand and stuff like that (Female, Pacific, 18 years).Parents will be more proactive about keeping their children healthy while they are younger. Plus, it will put a little more pressure on the younger people. Pressure might sometimes be bad but I think a lot of times a little bit of pressure is good to create the change but just don’t overload them with stress (Female, European, 18 years).

Ensuring adolescents are at the centre of future resources was viewed as fundamental. In order to improve understanding and awareness of DOHaD concepts, participants discussed the importance of resources and strategies designed with and for adolescents. They highlighted that traditional health media dissemination, such as pamphlets, booklets, or lengthy presentations, would not be suitable for capturing the attention of adolescents and that it would be important to ensure information was presented in an engaging way.

A resource booklet will be a waste of time—people will just throw them away. (We) need to present information in a way that will capture the attention of the adolescents (Female, European, 18 years).(The) presentation I got at school was not enjoyable. Presentations are a little too long. How people speak can get boring. It is really slow. They (health promoters) sound like they do not want to be there to deliver the message and they should consider the fun factor of stuff that could get us interested. Like fun activities (Male, Pacific, 18 years).

Instead, participants highlighted that video, social media, and other short form content could be used to help adolescents build their knowledge on, for example, the importance of the early life nutritional environment for later health. They emphasised that adolescents needed to be involved in this process and be able to see themselves in health promotion strategies or otherwise it would not be relatable. Participants discussed that the content of such approaches should focus on the concept of adolescent health and wellbeing rather than just talking about pregnancy.

I think most people my age are on social media. Make the information kind of fun and jokey…you just need people to watch it (such as) short form content. And have youth in the videos too because when you watch something and recognise yourself, you feel more welcome when there is someone in the video who is like you. If you have older people talking slowly, I don’t know if that will be effective (Female, European, 18 years).People our age don’t like to be flooded with information. I used to not appreciate it when people come in and always talk to us about pregnancy. We are sixteen! Don’t do it that way. It might feel a little out of place if you are talking about people younger than us or our children so make it about adolescents (and then) I think it would be interesting. You can make it fit quite nicely to the curriculum (Female, European, 18 years).

Overall, efforts to communicate DOHaD-related health promotion to adolescents was outlined as being pointless if the “why” aspect of the concept was not emphasised.

You (researchers/health promoters) have to explain it, then we will do what you guys are trying to stop us doing like vaping. You need to give the reasoning then it will make sense for me to stop bad behaviours (Female, Pacific, 16 years).

## 4. Discussion

### 4.1. Knowledge of DOHaD-Related Terminology

This study engaged a group of adolescents living in NZ in discussions about DOHaD to obtain their perspectives on current knowledge trends among adolescents and what might be done to improve understanding in this key age demographic. This study showed limited knowledge of DOHaD, the First 1000 days, and related concepts among adolescents. This finding is comparable to data reported from surveys examining awareness of DOHaD in first time mothers in Europe [23], school students and their parents in NZ [21], school students in Uganda [13], university students in Japan and NZ [10], and community members in the Cook Islands [22]. The limited knowledge of DOHaD was consistent with how adolescents in this study prioritised the importance of different health issues affecting their age group. Alcohol abuse and drug addiction were the two top issues that adolescents highlighted and are perhaps most known to this age group due to related health promotion activities in schools, churches, and communities [31]. However, as obesity and NCDs represent the most prevalent disease burdens in NZ, it is critical that adolescents are engaged in learning about such diseases, alongside potential early life prevention strategies to mitigate later life disease risk [32].

### 4.2. Understanding Different Life Course Windows for DOHaD Interventions

Adolescents did acknowledge that there are different life course windows for DOHaD interventions. Participants argued that the impacts of early life nutritional environment on health across the life course diminishes with each advancing life course. This argument highlights the knowledge gap in teenagers that requires addressing for the purpose of breaking the transgenerational cycle of disease. The disagreement adolescents in this study expressed with some DOHaD concepts was similar to findings from a study conducted by Oyamada and colleagues across Japan and NZ [10]. Disagreements with how important a mother’s health is before pregnancy, a father’s health before his partner becomes pregnant, and a child’s diet during the first two years of life for later life health can be representative of certain frames of reference that align with reductionist thinking [10]. Adolescents are social beings and as their social circle expands, they become exposed to different environments and people. This expansion of social circle allows adolescents to consider contrasting ideas and cultivates perspectives based on assumptions, conceptions, and expectations involving values, beliefs, and concepts [33].

### 4.3. Recognising That DOHaD-Related Information Needs to Be Accessible for Adolescents

The need for DOHaD information to be more widespread was recognised by adolescents in this study. This was underscored by the prioritisation of health issues that were deemed important to their age group. Although both drug addiction and alcohol abuse are important health issues, obesity is amongst the most common, and increasing, issue in communities in NZ. Thus, we would expect adolescents to prioritise obesity more than alcohol and drugs. However, this was not the case. Participants discussed that although there are resources available at school, adolescents would not be able to access the information they seek if they are not guided on what to look for. The emphasis on DOHaD-related information being taught in health-related subjects, which was compulsory for only two years of an adolescent’s high school journey in NZ, can be a barrier for information retainment in adolescence. The retaining of information in adolescence is a key motivational opportunity for the understanding of DOHaD concepts that can encourage sustainable behavioural change for later life [11]. Woods-Townsend and colleagues agreed that behavioural change is often the result of effective interventions; however, they further advise that sustained behavioural change requires additional personalised support for adolescents that transcends beyond education.

### 4.4. The Importance of Future Resources to Be Contextually Specific for Adolescents

Adolescents in this study identified a range of different avenues whereby DOHaD and related health information could be promoted, such as in clinics, schools, and homes. Participants suggested that the home environment is one’s first school and that it could hold more potential to improve literacy development compared to other settings [34]. However, they also acknowledged that this may not work for adolescents from families with low health literacy or who do not consider DOHaD and health related concepts as appropriate family or communal conversations [35]. Adolescents identified that health professionals could be one way to promote discussion however questioned whether they themselves knew enough about this topic. Oyamada and colleagues discussed that education may still be insufficient during health professional programmes to support healthcare professionals in understanding DOHaD principles and how it can be practically applied [10].

Another common avenue proposed by adolescents for health promotion initiatives was through the school-based environment. They discussed that initiatives needed to be tailored for the audience and discussed other school health programmes that had failed to engage adolescents. Participants discussed that health promotion could occur through the utilisation of social media platforms and videos that prioritise the health of adolescents in ways that captures their attention, engaging, empowering, and encouraging sustained behavioural change in this age group and cultivating better health outcomes for the future [13].

Resources that aspire to address the current gap in DOHaD awareness must recognise the importance of the adolescents themselves and their context [22]. This is in keeping with the context of health promotion in adolescents. To support DOHaD-related awareness in adolescents living in NZ, adolescents must be involved in research [36]. Adolescents need to collaborate with researchers to find ways to promote work in this field in a form that equips adolescents and empowers them to be catalysts of change in their society. The best way to go forward is to strengthen the modern education system to prepare young people to participate as critically engaged citizens and lifelong learners. This will not only support the present strive for collaborative research with adolescents but also negotiate interventions that may be useful for the future generation of parents as previously detailed by earlier research in this field [37].

### 4.5. Limitations and Future Directions

The findings from this study provide important perspectives on adolescents’ perceptions of DOHaD and strengthens the quantitative results we attained from a corresponding awareness survey. This study makes several original contributions to the field of DOHaD and adolescent health, particularly by providing a nuanced understanding of adolescents’ perspectives on health concepts that are critical to their developmental trajectories. Firstly, it fills a significant knowledge gap by exploring the awareness and perceptions of DOHaD principles among adolescents, a demographic often overlooked in DOHaD research, which traditionally focuses on early childhood or prenatal stages. By focusing on New Zealand adolescents, the study also adds a culturally and geographically specific dimension to the DOHaD discourse, highlighting unique societal and health system contexts that influence adolescent health understanding. Thus, a key strength of this research is the strong representation of Pacific ethnicities where some of the greatest health disparities exist. Furthermore, the employed qualitative approach allows for a rich, detailed exploration of adolescents’ views, offering insights into how these young individuals interpret and prioritise health information. These findings can inform the development of targeted educational interventions and public health strategies that resonate with adolescents, ultimately fostering a more informed and health-conscious youth population. By emphasising these aspects, the study not only enhances the existing body of DOHaD literature but also provides practical pathways for improving adolescent health outcomes through informed policymaking and community engagement. Although saturation of themes was reached in this study, we recognise the small sample size as a limitation. Of note, a recent systematic review by Hennink et al. has shown that, for such qualitative research, saturation and therefore reliability of data inferences can be reached at relatively small sample sizes, including at numbers reported in the current study [38]. All respondents from the previous survey who indicated interest in future research were followed up with. However, challenges related to participant recruitment including non-response, incorrect contact details, and short study timeframes limited the pool of adolescents. Despite the limitations, this study provided an important opportunity for adolescents to have their say and contribute to research in the DOHaD field. Future research could extend data collection to include more ethnic groups and widen the representation of geographic locations in NZ and beyond. Processes and strategies used in schools to provide DOHaD-related content is another avenue that could be explored in future studies. Although work in the school-based setting has highlighted that engagement with adolescents’ results in an improved understanding of basic DOHaD concepts and related behavioural changes over the shorter term [9], strategies are still required to ensure that such changes can be sustained longer term. Moreover, research could focus on contextually relevant and co-designed health promotion strategies by adolescents, for adolescents. The findings of this study have significant practical implications for developing targeted interventions and shaping public health policies in New Zealand. Given adolescents’ expressed desire for more accessible and engaging DOHaD-related information, policymakers should consider investigating the effectiveness of the current national curriculum in health and science education and the long-term effects in adolescents with regard to the understanding of basic DOHaD concepts and related behavioural changes. This aligns with recent research highlighting the importance of early life interventions in shaping long-term outcomes [39]. Furthermore, the adolescents suggest that digital platforms and social media could be effective channels for disseminating DOHaD information to adolescents. Public health agencies could partner with adolescents and popular social media influencers to co-develop engaging apps to deliver key health messages, an approach that has shown promise in recent youth-focused health campaigns [40]. Additionally, the identified knowledge gaps around the impact of paternal health on offspring outcomes suggest a need for public health campaigns that emphasise the role of both parents in early life health. This could involve community-based programmes targeting young adults of all genders, potentially reducing intergenerational health disparities [41]. By translating these findings into action, New Zealand has the opportunity to pioneer adolescent-centred DOHaD interventions, potentially setting global precedents in early life health promotion.

## 5. Conclusions

This qualitative study provides, for the first time, suggestive evidence as to why adolescents living in New Zealand disagreed with some DOHaD-related concepts. We found that adolescents had low awareness of DOHaD and DOHaD-related terminologies. Along with this, we report adolescents’ perspective of DOHaD concepts and suggestions for DOHaD-related information to be made accessible for adolescents. The results of this study offer significant practical implications for both public health interventions and future research in the field of adolescent health and DOHaD. These findings underscore the necessity for tailored health education programmes that incorporate DOHaD concepts aiming to increase awareness and understanding among adolescents. Such programmes could be integrated into the current school curricula, leveraging interactive and digital tools to engage students effectively. Additionally, the study highlights the importance of addressing diverse cultural perspectives, suggesting that future interventions should be culturally sensitive and inclusive to effectively reach all demographic groups within the New Zealand setting. For future research, longitudinal studies are recommended to explore how adolescents’ understanding of DOHaD concepts evolves over time and impacts their health behaviours. Furthermore, expanding the research to include a broader range of ethnic groups and geographical areas would provide a more comprehensive understanding of the research area. Investigating the role of digital media as a tool for health education and its effectiveness in conveying complex health concepts such as DOHaD could also be a promising avenue for future exploration. By addressing these areas, future studies can contribute to the development of more effective public health strategies that are informed by a deeper understanding of adolescent health perceptions and needs. Overall, future research should consider the adolescent voices presented in this study as a starting point to enabling the co-development of context-specific resources with adolescents, for adolescents. Co-designing with adolescents can encourage the uptake of information during this important developmental window.

## Figures and Tables

**Figure 1 ijerph-21-01556-f001:**
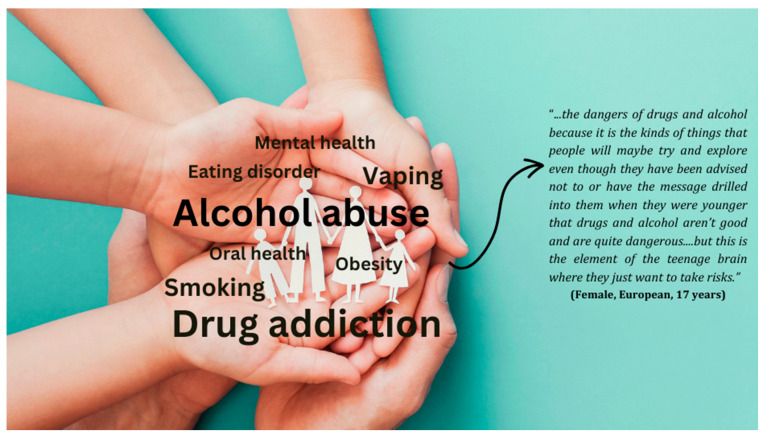
Health issues important for adolescence as identified by adolescents.

**Table 1 ijerph-21-01556-t001:** Characteristics of participants.

Characteristics.		N	%
Age	16 years	1	8
	17 years	5	42
	18 years	6	50
Gender	Female	8	67
	Male	4	33
Occupation	Employed	1	8
	High school student	10	84
	University student	1	8
Ethnicity	Pacific	7	58
	European	5	42
City	Auckland	9	76
	Wellington	1	8
	Christchurch	1	8
	Otago	1	8

**Table 2 ijerph-21-01556-t002:** Adolescent perspective on why survey respondents disagreed with DOHaD statements.

DOHaD Statements That Survey Respondents Disagreed with	Quotes From Adolescents
(i) A mother’s health BEFORE she becomes pregnant affects the health of the baby (foetus) during pregnancy	“I think it depends on how far the ages are apart. Say if a mother had health issues while she was a child herself, it will not affect her own child while she is pregnant. If she had health issues while she is carrying her baby within her then that will be more impactful than it happening before she becomes pregnant.” (Female, European, 18 years) “I think they (adolescents) would disagree because you do not exist before your mother gets pregnant.” (Female, European, 18 years) “I think that a lot of people would believe in that (anything that happens prior to being pregnant doesn’t really matter) because we always hear you do not smoke when you are pregnant, you do not drink when you are pregnant but they do forget that BEFORE is important as well.” (Female, European, 18 years)
(ii) The father’s health BEFORE his partner becomes pregnant affects the health of the baby (foetus) during pregnancy	“I think they would disagree because the father isn’t the one who is carrying the baby.” (Female, European, 18 years) “I think they might only be thinking about the certain effects and stuff…having the other parent not being held in good health situation or be seriously ill can take a toll and be stressful on the parent that is actually carrying the baby which can have other bad things happen due to physical impacts of stress.” (Female, European, 17 years) “They might think that the womb environment may have a much bigger impact than the father’s lungs impacting the health of the baby. It is just a lot closer contact to the baby. I would expect the mother’s health to be more important than the father’s health.” (Female, European, 18 years) “He is not conceiving the child. He is just doing his business so the rest does not matter anymore. I think motherhood is just, people often forget the paternal figure during the pregnancy and early life stages into the child’s life.” (Female, European, 18 years) “Because the baby is not growing inside of the father….there is not much information on how you can explain to a child who doesn’t really understand how a child is made…..some adolescents do not know that a child carries half of their father’s chromosomes so whatever he carries could affect the future child.” (Female, Pacific, 17 years)
(iii) The food that a child is fed during the first two years of life affects their health throughout childhood and adulthood	“I think they would disagree because it is only two years. They do not see the important in for example breast milk compared to formula. Your lifetime is however long and you have so many opportunities to be healthy and change your diet.” (Female, Other, 18 years) “They could think that because may be after being fed whatever your parents feed you for two years then their lifestyle changed and that it did not really affect them in the future.” (Female, Pacific, 17 years)

## Data Availability

Data are contained within the article.

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
