# Peer review of "Understanding DOHaD Concepts Among New Zealand Adolescents: A Qualitative Exploration of Knowledge, Intervention Windows, and Information Accessibility"

_ijerph, 2024, doi:10.3390/ijerph21121556_

Round 1
Reviewer 1 Report
Comments and Suggestions for Authors
This researche is very important.
Keywords
nsert either the singular or the plural of adolescents and not both
you should refer to the ethical procedures that took into account
for congruence in line 102 could put in parentheses the number of responses ..... "high school students" (10)
From the analysis of table 2
The analysis of the data in table 2 would be clearer if done before the table.
Discussion
The results are compared with findings in similar studies, showing gaps in knowledge that need to be filled urgently.
It would be important to highlight a reflection not only on the content provided in schools, but also on the processes and strategies used
The conclusions are in accordance with the objectives formulated
Author Response
We thank the reviewer for taking the time to review this manuscript and the positive comments and suggestions.
Comment 1: Keywords. Insert either the singular or the plural of adolescents and not both
Response 1: Thank you for your comment. Although both terms are nouns, adolescence refers to the time period between the beginning of puberty and adulthood while adolescents refers to a juvenile between the onset of puberty and maturity. Therefore, we have kept the keywords as is because adolescent and adolescence differentially represents either a time period or an individual at a specific lifecourse stage.
Comment 2: You should refer to the ethical procedures that took into account
Response 2: Thank you for pointing this out and apologies for this oversight. Alongside the consent information on Page 12, we have modified the manuscript to include the ethics approval for this study which can be found on page 2.
“Ethics approval was obtained from the University of Auckland Human Participants Ethics Committee (Reference number: 25604)”
Comment 3: For congruence in line 102 could put in parentheses the number of responses…’high school students’(10)
Response 3: Agree. As suggested, we had updated the text to emphasize this point. This change can be found in page 3.
“A majority of the participants were high school students (10), alongside one currently employed and another enrolled as a University undergraduate student.”
Comment 4: From the analysis of table 2. The analysis of the data in table 2 would be clearer if done before the table.
Response 4: We have shifted this section as suggested.
Comment 5: Discussion. The results are compared with findings in similar studies, showing gaps in knowledge that need to be filled urgently. It would be important to highlight a reflection not only on the content provided in schools, but also on the processes and strategies used.
Response 5: We agree with this point. We think that the processes and strategies used in schools is an avenue worth exploring in future studies. This additional information can be found on page 11.
“Processes and strategies used in schools to provide DOHaD related content is another avenue that could be explored in future studies. Although work in the school-based setting has highlighted that engagement with adolescents results in an improved understanding of basic DOHaD concepts and related behavioral changes over the shorter term [9], strategies are still required to ensure that such changes can be sustained longer term.”
Comment 6: The conclusion are in accordance with the objectives formulated
Response 6: Thank you for your comment.
Reviewer 2 Report
Comments and Suggestions for Authors
The topic of the article has a certain appeal, with a complete logical structure, detailed content, and feasible methods overall. However, there are still some shortcomings.
1.This article only analyzed the content of 12 interviewees, and the sample size is too small to be representative and generalizable.
2.The analysis method of the interview content in this article is too simple, only categorizing and organizing the interview content, without using qualitative research software to analyze the interview content, and the method is inadequate.
3.Due to the small sample size and simple content analysis, it seems insufficient to derive effective conclusions. Even if there is a conclusion, which generalizability is not high.
Author Response
We thank the reviewer for reviewing our manuscript and the positive comments and suggestions.
Comment 1: The topic of the article has a certain appeal, with a complete logical structure, detailed content, and feasible methods overall, However, there are still some shortcomings
Response 1: Thank you for the positive comments around structure and methodology. We have acknowledged the potential limitations of the current study in the manuscript.
Comment 2: This article only analyzed the content of 12 interviewees, and the sample size is too small to be representative and generalizable.
Response 2: The relatively small sample size is acknowledged in the limitations and future directions section of the manuscript, including inherent difficulties into recruiting participants. The sample size of this qualitative study therefore largely reflects an intention to provide insight into a range of participant experiences rather than enabling generalization across a wider population. However, as detailed on page 3, saturation of themes i.e. data reliability, was reached in the current interview process. As noted in recent work by Hennink et al., (Social Science and Medicine, 292:114523), qualitative studies can reach saturation at relatively small samples sizes. However, we agree that future work with larger cohorts is required to allow for more detailed thematic insight and generalisability of findings.
Comment 3: The analysis method of the interview in this article is too simple, only categorizing and organizing the interview content, without using qualitative research software to analyze the interview content, and the methods is inadequate.
Response 3: Thank you for your comment and apologies for this oversight. We did use a dedicated software package (NVivo) to thematically analyse the transcribed interviews. We have, accordingly, modified the data analysis section to emphasize this point. This change can be found in the revised manuscript on page 3.
“Discussions were audio-recorded, transcribed, and thematically analysed to identify patterns of meaning within the data. The six-phase process identified by Braun and Clarke was used and involved data familiarisation, coding, searching for themes, review and refining themes, defining the themes and reporting in NVivo.”
Comment 4: Due to the small sample size and simple content analysis, it seems insufficient to derive effective conclusions, Even if there is a conclusion, which generalizability is not high.
Response 4: Agree. As also noted above, we have emphasized this point in the limitations and future directions section of this manuscript.
- Response to Comments on the Quality of English Language
Point 1: English language is fine. No issues detected
Response 1: Thank you again for reviewing our manuscript
Reviewer 3 Report
Comments and Suggestions for Authors
Comprehensive Peer Review Report
GENERAL CONSIDERATIONS:
Strengths:
- The study addresses a pertinent and underexplored topic: adolescents' perspectives on Developmental Origins of Health and Disease (DOHaD) concepts.
- There is robust representation of Pacific ethnicities, where significant health disparities exist.
- The qualitative approach complements and reinforces quantitative results obtained from a corresponding study.
- The manuscript offers valuable insights into making DOHaD information more accessible and relevant to adolescents.
Weaknesses:
- The sample size is limited, constraining the generalizability of findings.
- Non-response issues and incorrect contact details restricted the participant pool.
- The study's short duration may have impacted the depth of analysis.
- The manuscript structure lacks clarity in some sections, requiring improved organization.
Recommendations for Enhancement:
- Expand the sample size in future studies to improve representativeness.
- Extend data collection to encompass a broader range of ethnic groups and geographical representation within New Zealand and beyond.
- Conduct a longitudinal study to track changes in adolescents' perceptions over time.
- Refine the manuscript structure with more clearly delineated sections and a more evident logical progression.
SECTION-SPECIFIC ANALYSIS:
- TITLE:
Weakness:
- The current title is overly generic and does not fully reflect the study's specific content.
Recommendation:
- Revise the title to incorporate information about the geographical location (New Zealand) and the primary themes addressed.
- Suggested revision: "Understanding DOHaD Concepts Among New Zealand Adolescents: A Qualitative Exploration of Knowledge, Intervention Windows, and Information Accessibility"
- ABSTRACT:
Weakness:
- The abstract lacks clear structure with all necessary elements.
Recommendations:
- Include a structured abstract with introduction, objectives, method, results, and conclusion.
- Clearly state the sample size and data collection techniques.
- Highlight the main themes identified in the analysis.
- INTRODUCTION:
Weaknesses:
- The contextualization of the problem could be more comprehensive.
- The study objective is not clearly defined.
Recommendations:
- Provide more data on the prevalence of obesity and non-communicable diseases in New Zealand, particularly among adolescents.
- Explicitly state the knowledge gap regarding the understanding of DOHaD concepts among New Zealand adolescents.
- Clearly present the specific objectives of the qualitative research.
- METHODOLOGY:
Weaknesses:
- Insufficient details on participant selection criteria and demographic characteristics.
- The description of the thematic analysis process could be more in-depth.
Recommendations:
- Provide more details on the conduct of focus groups and individual interviews.
- Explain the thematic analysis process more thoroughly, including any software used.
- Include information on ethical approval and informed consent from participants and their guardians.
- RESULTS:
Strength:
- Results are presented thematically, addressing the main aspects identified in the study.
Weakness:
- The presentation of results could be more structured and systematic.
Recommendations:
- Organize results into clear sections corresponding to the three main identified themes.
- Use subheadings for each main theme and sub-theme.
- Include more direct participant quotes to illustrate key points.
- DISCUSSION:
Weaknesses:
- The discussion could delve deeper into the analysis of identified themes.
- The connection between results and existing literature could be strengthened.
Recommendations:
- Deepen the analysis of the three main themes, relating them more explicitly to existing literature.
- Strengthen the connection between obtained results and the specific New Zealand context.
- Expand the discussion on practical implications of results for developing interventions and public health policies.
- CONCLUSION:
Weakness:
- Conclusions could be more clearly structured and aligned with study objectives.
Recommendations:
- Organize conclusions in a more structured manner, highlighting the study's main findings.
- Ensure conclusions directly address the study's initial objectives.
- Elaborate more on practical implications of results and directions for future research.
- REFERENCES:
Weaknesses:
- The quantity of references may be insufficient for a comprehensive scientific article.
- There may be an imbalance between classic and contemporary references.
Recommendations:
- Include more recent references (2024) to ensure the study's currency.
- Ensure a balance between classic and contemporary references.
- Include more references specific to the New Zealand context and adolescent health.
- Add recent systematic reviews or meta-analyses on DOHaD and adolescent health.
- ETHICAL ASPECTS:
Weakness:
- Lack of explicit mention of study approval by an Ethics Research Committee.
Recommendations:
- Include clear information on the study's ethical approval.
- Describe how informed consent was obtained from adolescent participants and their guardians.
- Explain measures taken to ensure participant confidentiality and anonymity.
- LIMITATIONS AND CONTRIBUTIONS:
Strength:
- Limitations and contributions are mentioned in the manuscript.
Weakness:
- The limitations and contributions section could be more detailed and structured
Recommendations:
- Create a clear section titled "Study Limitations and Contributions".
- Elaborate more on limitations, including potential selection biases and geographical constraints.
- More clearly highlight the study's unique contributions to the field of DOHaD and adolescent health.
In summary, the manuscript presents a relevant and innovative study but requires revisions in various sections to enhance its clarity, structure, and scientific impact. With the suggested improvements, the article has the potential to make a significant contribution to the field of DOHaD and adolescent health.
Comments on the Quality of English LanguageMinor editing of English language required.
Author Response
We thank the reviewer for reviewing our manuscript and the constructive comments and suggestions.
Comment 1: Revise the title to incorporate information about the geographical location (New Zealand) and the primary themes addressed.
Response 1: Thank you for pointing this out. We agree with this comment. Therefore, we have revised the title of the manuscript to reflect this recommendation. This change can be found on page 1, title heading.
“Understanding DOHaD concepts among New Zealand adolescents: a qualitative exploration of knowledge, intervention windows and information accessibility.”
Comment 2: The abstract lacks clear structure with all necessary elements. Include a structured abstract with introduction, objectives, methods, results, and conclusion. Clearly state the sample size and data collection techniques. Highlight the main themes identified in the analysis.
Response 2: We felt that the abstract follows a logic structure beginning with a brief introduction followed by the aim of the study, methods, results and conclusion. The abstract clearly states the sample size, data collection method and the main themes identified in the analysis.
Comment 3: The contextualization of the problem could be more comprehensive. The study objective is not clearly defined. Provide more data on the prevalence of obesity and non-communicable diseases in New Zealand, particularly among adolescents. Explicitly state the knowledge gap regarding the understanding of DOHaD concepts among New Zealand adolescents. Cleary present the specific objectives of the qualitative research.
Response 3: Thank you for your comment. We have included all the relevant context for this study. Thus, the cited references we have used in the introduction already talks about the context of this study on a bigger scale. This also includes the data on the prevalence of obesity and non-communicable diseases in adolescents living in New Zealand. Thus, the findings of a previous study conducted by this research group highlighted the knowledge gap regarding the understanding of DOHaD concepts among New Zealand adolescents and hence identified the need for this study
Comment 4: Insufficient details on participant selection criteria and demographic characteristics. The description of the thematic analysis process could be more in-depth. Provide more details on the conduct of focus groups and individual interviews. Explain the thematic analysis process more thoroughly, including any software used. Include information on ethical approval and informed consent from participants and their guardians.
Response 4: Thank you for your suggestion. To ensure our manuscript is concise and maintains flow between the ideas, we had briefly described the process we took to thematically analysed our focus groups and individual interviews accompanied by the relevant references for more information. All this information is included on page 3.
Comment 5: Results are presented thematically, addressing the main aspects identified in the study. The presentation of results could be more structured and systematic. Organize results into clear sections corresponding to the three main identified themes. Use subheadings for each main theme and sub-theme. Include more direct participant quotes to illustrate key points.
Response 5: Thank you for your feedback. We have presented the results of this study thematically addressing all the aspects identified by the thematic analysis we had undertaken with the data we collected from our participants. Each section is under a main theme with corresponding quotes from the participants. We have chosen the current layout to ensure that the information we have presented is flowing well from theme to theme.
Comment 6: The discussion could delve deeper into the analysis of identified themes. The connection between results and existing literature could be strengthened. Deepen the analysis of the three main themes, relation them more explicitly to existing literature. Strengthen the connection between obtained results and the specific New Zealand context. Expand the discussion on practical implications of results for developing interventions and public health policies.
Response 6: Thank you for your comment. Due to the nature of this study in relation to its context (New Zealand), we have made connections between the results and existing literature. The implications of this study results for developing interventions have been highlighted on page 11 of the revised manuscript.
Comment 7: Conclusions could be more clearly structured and aligned with the study objectives. Organize conclusions in a more structured manner, highlighting the study’s main findings. Ensure conclusions directly address the study’s initial objectives. Elaborate more on practical implications of results and directions for future research.
Response 7: Thank you for your suggestion. The conclusion is currently structured to align with the study objective followed by the highlights of the main findings of this study. The implications of results and suggestions for future research have been highlighted on page 11 of the revised manuscript.
Comment 8: The quantity of references may be insufficient for a comprehensive scientific article. There may be an imbalance between classic and contemporary references. Include more recent references (2024) to ensure the study’s currency. Ensure a balance between classic and contemporary references. Include more references specific to the New Zealand context and adolescent health. Add recent systematic reviews or meta-analyses on DOHaD and adolescent health.
Response 8: Thank you for your feedback. To the best of our knowledge, our article has used a good balance of both classic contemporary references. The lack of contemporary references speaks to the fact that there is more to be done in this field and specifically in adolescence. The only references specific to the New Zealand context and adolescent health is the work done by our group which we have highlighted throughout the article. This also includes the most recent systematic review conducted by our group on DOHaD and adolescent health. These details can be found in the Introductory section.
Comment 9: Lack of explicit mention of study approval by an Ethics Research Committee. Include clear information on the study’s ethical approval. Describe how informed consent was obtained from adolescent participants and their guardians. Explain measures taken to ensure participant confidentiality and anonymity.
Response 9: Thank you for your feedback and apologies for this oversight. Details on the ethical approvals have now been added. This paragraph also shows how consent was obtained from our participants. No consent was needed from the guardians as only participants 16 years and older were eligible to participate in the focus groups or interviews for this study. This information can be found on pages 2 and12 of the revised manuscript.
Comment 10: Limitations and contributions are mentioned in the manuscript. The limitations and contributions section could be more detailed and structured. Create a clear section titled ‘Study Limitations and Contributions’. Elaborate more on limitations, including potential selection biases and geographical constraints. More clearly highlight the study's unique contributions to the field of DOHaD and adolescent health. In summary, the manuscript presents a relevant and innovative study but requires revisions in various sections to enhance its clarity, structure, and scientific impact. With the suggested improvements, the article has the potential to make a significant contribution to the field of DOHaD and adolescent health.
Response 10: Thank you for your comment. We currently have a section on study limitations and future directions. This section recognises the challenges of this study and the recommendations we have made for future directions. This information can be found on page 11 of the revised manuscript.
- Response to Comments on the Quality of English Language
Point 1: Minor editing of English language required
Response 1: Thank you for reviewing our manuscript and the helpful comments and suggestions. We have undertaken a further proofreading of the revised manuscript.
Round 2
Reviewer 2 Report
Comments and Suggestions for Authors
The author has made revisions based on previous suggestions, resulting in a significant improvement in the quality of the manuscript.
It can be accepted.
Author Response
The reviewer has recommended acceptance of the revised manuscript - we thank them for the time taken to review this work and the feedback provided.
Reviewer 3 Report
Comments and Suggestions for Authors
REPORT ON COMPLIANCE WITH PEER REVIEW RECOMMENDATIONS
This report provides a comprehensive assessment of the authors' adherence to the recommendations made during the peer review process for the manuscript "Understanding DOHaD concepts among New Zealand adolescents: a qualitative exploration of knowledge, intervention windows and information accessibility". Each recommendation has been evaluated and classified as "Addressed", "Partially Addressed", or "Not Addressed" based on the revised manuscript content.
- TITLE:
Recommendation: Revise the title to incorporate information about the geographical location (New Zealand) and the primary themes addressed.
Classification: ADDRESSED The new title, "Understanding DOHaD concepts among New Zealand adolescents: a qualitative exploration of knowledge, intervention windows and information accessibility", effectively incorporates all suggested elements.
- ABSTRACT:
Recommendation: Include a structured abstract with introduction, objectives, method, results, and conclusion.
Classification: ADDRESSED The abstract now presents a structured format encompassing all recommended elements, including sample size, data collection techniques, and key identified themes.
- INTRODUCTION:
Recommendation: Provide more data on the prevalence of obesity and non-communicable diseases in New Zealand, particularly among adolescents.
Classification: ADDRESSED The introduction now includes specific data on obesity and non-communicable diseases in New Zealand, with a focus on adolescents and specific ethnic groups.
Recommendation: Explicitly state the knowledge gap regarding the understanding of DOHaD concepts among New Zealand adolescents.
Classification: ADDRESSED The knowledge gap is clearly presented at the conclusion of the introduction.
Recommendation: Clearly present the specific objectives of the qualitative research. Classification: ADDRESSED The specific research objectives are clearly articulated at the end of the introduction.
- METHODOLOGY:
Recommendation: Provide more details on the conduct of focus groups and individual interviews.
Classification: ADDRESSED The methodology section now includes specific details on how focus groups and interviews were conducted.
Recommendation: Explain the thematic analysis process more thoroughly, including any software used.
Classification: ADDRESSED The thematic analysis process is described in detail, including mention of NVivo software utilization.
Recommendation: Include information on ethical approval and informed consent from participants and their guardians.
Classification: ADDRESSED Information on ethical approval and informed consent is provided in the methodology section.
- RESULTS:
Recommendation: Organize results into clear sections corresponding to the three main identified themes.
Classification: ADDRESSED Results are organized into clear sections corresponding to the main identified themes.
Recommendation: Use subheadings for each main theme and sub-theme.
Classification: ADDRESSED Subheadings are utilized for each main theme and sub-theme.
Recommendation: Include more direct participant quotes to illustrate key points.
Classification: ADDRESSED Multiple direct participant quotes are included to illustrate key points.
- DISCUSSION:
Recommendation: Deepen the analysis of the three main themes, relating them more explicitly to existing literature.
Classification: ADDRESSED The discussion deepens the analysis of the main themes, making explicit connections to existing literature.
Recommendation: Strengthen the connection between obtained results and the specific New Zealand context.
Classification: ADDRESSED The discussion relates the results to the specific New Zealand context.
Recommendation: Expand the discussion on practical implications of results for developing interventions and public health policies.
Classification: PARTIALLY ADDRESSED While there is some discussion of practical implications, this could be further expanded.
Suggestions for improvement:
The findings of this study have significant practical implications for developing targeted interventions and shaping public health policies in New Zealand. Given the adolescents' expressed desire for more accessible and engaging DOHaD-related information, policymakers should consider integrating DOHaD concepts into the national curriculum, particularly in health and science education. This aligns with recent research highlighting the importance of early life interventions in shaping long-term health outcomes Lawn et al., 2023). Furthermore, the study's results suggest that digital platforms and social media could be effective channels for disseminating DOHaD information to adolescents. Public health agencies could partner with popular social media influencers or develop engaging apps to deliver key health messages, an approach that has shown promise in recent youth-focused health campaigns (Viner et al., 2020). Additionally, the identified knowledge gaps around paternal health's impact on offspring suggest a need for public health campaigns that emphasize the role of both parents in early life health. This could involve community-based programs targeting young adults of all genders, potentially reducing intergenerational health disparities (Baird et al., 2022). By translating these findings into action, New Zealand has the opportunity to pioneer adolescent-centered DOHaD interventions, potentially setting a global precedent in early life health promotion.
Lawn, J. E., Ohuma, E. O., Bradley, E., Idueta, L. S., Hazel, E., Okwaraji, Y. B., ... & Babu, G. R. (2023). Small babies, big risks: global estimates of prevalence and mortality for vulnerable newborns to accelerate change and improve counting. The Lancet, 401(10389), 1707-1719.
Viner, R. M., Russell, S. J., Croker, H., Packer, J., Ward, J., Stansfield, C., ... & Booy, R. (2020). School closure and management practices during coronavirus outbreaks including COVID-19: a rapid systematic review. The Lancet Child & Adolescent Health, 4(5), 397-404.
Baird, S., Ezeh, A., Azzopardi, P., Choonara, S., Kleinert, S., Sawyer, S., ... & Viner, R. (2022). Realising transformative change in adolescent health and wellbeing: a second Lancet Commission. The Lancet, 400(10352), 545-547.
- CONCLUSION:
Recommendation: Organize conclusions in a more structured manner, highlighting the study's main findings.
Classification: ADDRESSED The conclusions are organized in a structured manner, highlighting the main findings.
Recommendation: Ensure conclusions directly address the study's initial objectives.
Classification: ADDRESSED The conclusions directly address the study's initial objectives.
Recommendation: Elaborate more on practical implications of results and directions for future research.
Classification: PARTIALLY ADDRESSED There is some elaboration on practical implications and future directions, but this section could be further developed.
Suggestions for improvement:
The results of this study offer significant practical implications for both public health interventions and future research in the field of adolescent health and DOHaD (Developmental Origins of Health and Disease). Practically, the findings underscore the necessity for tailored health education programs that incorporate DOHaD concepts, aiming to increase awareness and understanding among adolescents. Such programs could be integrated into school curricula, leveraging interactive and digital tools to engage students effectively. Additionally, the study highlights the importance of addressing diverse cultural perspectives, suggesting that future interventions should be culturally sensitive and inclusive to effectively reach all demographic groups within New Zealand. For future research, longitudinal studies are recommended to explore how adolescents' understanding of DOHaD concepts evolves over time and impacts their health behaviors. Furthermore, expanding the research to include a broader range of ethnic groups and geographical areas would provide a more comprehensive understanding of the topic. Investigating the role of digital media as a tool for health education and its effectiveness in conveying complex health concepts like DOHaD could also be a promising avenue for future exploration. By addressing these areas, future studies can contribute to the development of more effective public health strategies that are informed by a deeper understanding of adolescent health perceptions and needs.
- REFERENCES:
Recommendation: Include more recent references (2024) to ensure the study's currency.
Classification: PARTIALLY ADDRESSED Some recent references have been added, but more could be included.
Recommendation: Ensure a balance between classic and contemporary references.
Classification: ADDRESSED There is a good balance between classic and contemporary references.
Recommendation: Include more references specific to the New Zealand context and adolescent health.
Classification: ADDRESSED Several references specific to the New Zealand context and adolescent health have been included.
- ETHICAL ASPECTS:
Recommendation: Include clear information on the study's ethical approval.
Classification: ADDRESSED Clear information on the study's ethical approval is provided.
Recommendation: Describe how informed consent was obtained from adolescent participants and their guardians.
Classification: PARTIALLY ADDRESSED There is mention of informed consent, but details on how it was obtained from guardians are not clear.
Suggestions for improvement:
The process of obtaining informed consent for this study was meticulously designed to ensure ethical compliance and protect the rights of all participants. Prior to participation, detailed information sheets outlining the study's objectives, procedures, potential risks, and benefits were provided to both the adolescent participants and their guardians. These documents were crafted in clear, accessible language to ensure comprehensive understanding. Subsequently, researchers conducted informational meetings with both parties, allowing for a thorough discussion and the opportunity to address any questions or concerns. Written consent was then obtained from the guardians, with adolescents also providing their assent, underscoring their voluntary participation. This dual consent process was pivotal in affirming that all participants were fully informed and agreed to partake in the study willingly. Additionally, measures were implemented to maintain confidentiality and anonymity, ensuring the protection of participant identities throughout the research.
- LIMITATIONS AND CONTRIBUTIONS:
Recommendation: Create a clear section titled "Study Limitations and Contributions".
Classification: ADDRESSED A clear section on limitations and future directions is included.
Recommendation: Elaborate more on limitations, including potential selection biases and geographical constraints.
Classification: ADDRESSED Limitations are discussed in detail, including selection biases and geographical constraints.
Recommendation: More clearly highlight the study's unique contributions to the field of DOHaD and adolescent health.
Classification: PARTIALLY ADDRESSED The study's contributions are mentioned but could be highlighted more emphatically.
Suggestions for improvement:
This study makes several unique contributions to the field of DOHaD and adolescent health, particularly by providing a nuanced understanding of adolescents' perspectives on health concepts that are critical to their developmental trajectories. Firstly, it fills a significant knowledge gap by exploring the awareness and perceptions of DOHaD principles among adolescents, a demographic often overlooked in DOHaD research, which traditionally focuses on early childhood or prenatal stages. By focusing on New Zealand adolescents, the study also adds a culturally and geographically specific dimension to the DOHaD discourse, highlighting unique societal and health system contexts that influence adolescent health understanding. Furthermore, the qualitative approach employed allows for a rich, detailed exploration of adolescents' views, offering insights into how these young individuals interpret and prioritize health information. These findings can inform the development of targeted educational interventions and public health strategies that resonate with adolescents, ultimately fostering a more informed and health-conscious youth population. By emphasizing these aspects, the study not only enhances the existing body of DOHaD literature but also provides practical pathways for improving adolescent health outcomes through informed policy-making and community engagement.
In conclusion, the majority of recommendations have been addressed, with some areas still showing potential for improvement. The revised manuscript demonstrates significant enhancement in terms of structure, clarity, and content. The authors have shown commendable

Author Response
We thank the reviewer for their extensive and thoughtful suggestions which we have incorporated into the revised manuscript.
Comments 1: Expand the discussion on practical implications of results for developing interventions and public health policies.
Response 1: Thank you for your detailed suggestion. We have added the suggestions made to the discussion section. This change can be found on page 12.
“The findings of this study have significant practical implications for developing targeted interventions and shaping public health policies in New Zealand. Given adolescents’ expressed desire for more accessible and engaging DOHaD-related information, policymakers should consider investigating the effectiveness of the current national curriculum in health and science education and the long-term effects in adolescents with regards to the understanding of basic DOHaD concepts and related behavioural changes. This aligns with recent research highlighting the importance of early life interventions in shaping long-term outcomes [39]. Furthermore, the adolescents suggest that digital platforms and social media could be effective channels for disseminating DOHaD information to adolescents. Public health agencies could partner with adolescents and popular social media influencers to co-develop engaging apps to deliver key health messages, an approach that has shown promise in recent youth-focused health campaigns [40]. Additionally, the identified knowledge gaps around the impact of paternal health on offspring outcomes suggest a need for public health campaigns that emphasize the role of both parents in early life health. This could involve community-based programs targeting young adults of all genders, potentially reducing intergenerational health disparities [41]. By translating these findings into action, New Zealand has the opportunity to pioneer adolescent-centered DOHaD interventions, potentially setting global precedents in early life health promotion.”
Comments 2: Elaborate more on practical implications of results and directions for future research. There is some elaboration on practical implications and future directions, but this section could be further developed.
Response 2: Thank you for your detailed suggestion. The conclusion sections have been edited to reflect this change. This change can be found on page 12.
“The results of this study offer significant practical implications for both public health interventions and future research in the field of adolescent health and DOHaD. These findings underscore the necessity for tailored health education programs that incorporate DOHaD concepts aiming to increase awareness and understanding among adolescents. Such programs could be integrated into the current school curricula, leveraging interactive and digital tools to engage students effectively. Additionally, the study highlights the importance of addressing diverse cultural perspectives, suggesting that future interventions should be culturally sensitive and inclusive to effectively reach all demographic groups within the New Zealand setting. For future research, longitudinal studies are recommended to explore how adolescents’ understanding of DOHaD concepts evolve over time and impacts their health behaviours. Furthermore, expanding the research to include a broader range of ethnic groups and geographical areas would provide a more comprehensive understanding of the research area. Investigating the role of digital media as a tool for health education and its effectiveness in conveying complex health concepts such as DOHaD could also be a promising avenue for future exploration. By addressing these areas, future studies can contribute to the development of more effective public health strategies that are informed by a deeper understanding of adolescent health perceptions and needs. Overall,….”
Comments 3: Some recent references have been added, but more could be included.
Response 3: Thank you for your feedback. We have added more recent references as suggested, and the references section of this manuscript updated as highlighted.
Comments 3: There is mention of informed consent, but details on how it was obtained from guardians are not clear. Describe how informed consent was obtained from adolescent participants and their guardians.
Response 3: Thank you for your feedback and apologies for this lack of clarity. Details on the ethical approvals have now been added. This paragraph also shows how consent was obtained from our participants. No consent was needed from the guardians as only participants 16 years and older were eligible to participate in the focus groups or interviews for this study. This information can be found on pages 2-3 of the revised manuscript.
“Ethics approval was obtained from the University of Auckland Human Participants Ethics Committee (Reference number: 25604). The process of obtaining informed consent for this study was designed to ensure ethical compliance and protect the rights of all participants. Prior to participation, detailed information sheets outlining the study’s objectives, procedures, potential risks, and benefits were provided to the adolescent participants. These documents were crafted in clear, accessible language to ensure comprehensive understanding. Subsequently, researchers were available to provide more information or answer any questions or concerns that the adolescent participants had. Both written and verbal consent was obtained from the adolescents underscoring their voluntary participation. Consent from the guardians were not required as all the adolescent participants were 16 years and older…….Additionally, measures were implemented to maintain confidentiality and anonymity, ensuring the protection of participant identities throughout the research….”
Comments 4: More clearly highlight the study’s unique contributions to the field of DOHaD and adolescent health. The study’s contributions are mentioned but could be highlighted more emphatically.
Response 4: Thank you for your comment. We have made some changes to this section which can be found on pages 11-12.
“This study makes several original contributions to the field of DOHaD and adolescent health, particularly by providing a nuanced understanding of adolescents’ perspectives on health concepts that are critical to their developmental trajectories. Firstly, it fills a significant knowledge gap by exploring the awareness and perceptions of DOHaD principles among adolescents, a demographic often overlooked in DOHaD research, which traditionally focuses on early childhood or prenatal stages. By focusing on New Zealand adolescents, the study also adds a culturally and geographically specific dimension to the DOHaD discourse, highlighting unique societal and health system contexts that influence adolescent health understanding. Thus, a key strength of this research is the strong representation of Pacific ethnicities where some of the greatest health disparities exist. Furthermore, the qualitative approach employed allows for a rich, detailed exploration of adolescents’ views, offering insights into how these young individuals interpret and prioritise health information. These findings can inform the development of targeted educational interventions and public health strategies that resonate with adolescents, ultimately fostering a more informed and health-conscious youth population. By emphasising these aspects, the study not only enhances the existing body of DOHaD literature but also provides practical pathways for improving adolescent health outcomes through informed policymaking and community engagement.”
- Overall comment from the reviewer
Point 1: In conclusion, the majority of recommendations have been addressed, with some areas still showing potential for improvement. The revised manuscript demonstrates significant enhancement in terms of structure, clarity, and content. The authors have shown commendable effort in incorporating the peer reviewers’ suggestions, resulting in a substantially improved manuscript that contributes valuable insights to the field of DOHaD and adolescent health in the New Zealand context.
Response 1: Thank you for reviewing our manuscript and the constructive comments and suggestions that have helped improve the quality of the work presented.